# Simulation-Based Assessment of Subsurface Drip Irrigation Efficiency for Crops Grown in Raised Beds

Vsevolod Bohaienko [1,*], Mykhailo Romashchenko [2], Andrii Shatkovskyi [3] and Maksym Scherbatiuk [3]

1 VM Glushkov Institute of Cybernetics of NAS of Ukraine, 03187 Kyiv, Ukraine
2 Research Sector, Kyiv Agrarian University, 03022 Kyiv, Ukraine; mi.romashchenko@gmail.com
3 Institute of Water Problems and Land Reclamation, National Academy of Agrarian Sciences of Ukraine, 03022 Kyiv, Ukraine; andriy-1804@ukr.net (A.S.); sherbatuykmaksim@gmail.com (M.S.)
* Correspondence: sevab@ukr.net

**Abstract:** This paper considers the application of a scenario simulation technique to assess subsurface drip irrigation system efficiency while using it to irrigate crops grown under raised bed technology. For simulating purposes, we use a model based on the two-dimensional Richards equation stated in terms of water head in a curvilinear domain. Solutions to problems are obtained using a finite-difference scheme with dynamic time step change. Using the data from pressure measurements obtained while growing potatoes on sandy loess soil in production conditions, we performed a calibration of the model using the particle swarm optimization algorithm. Further, the accuracy of the model was tested and average absolute errors in the range from 3.16 to 5.29 kPa were obtained. Having a calibrated model, we performed a series of simulations with different irrigation pipeline placements determining the configuration under which water losses are minimal. The simulated configuration, under which infiltration losses were minimal, was the installation of pipelines under the raised bed at the depth of 10 cm below the soil surface. The results confirm that the applied technique can be used for decision-making support while designing subsurface drip irrigation systems combined with raised bed growing technology.

**Keywords:** simulation; moisture transport; subsurface drip irrigation; raised bed technology





## 1. Introduction

The increasingly widespread use of drip irrigation, especially its subsurface variety, has partly developed in combination with specific technologies for growing agricultural crops. Such cases include the use of subsurface drip irrigation (SDI) for watering potatoes, carrots, onions, berries and other crops cultivated using raised bed technology. Here, while implementing technological processes of irrigation, it is necessary to solve a number of problems related to the determination of the parameters of irrigation and water supply regimes, the application of which ensures the maximum economic effect with minimal capital expenses and water consumption.

In this context, the efficiency of SDI systems is limited due to the lack of well-established methods for accurate determination in the specific conditions of a particular farm of a systems' design parameters (first, the depth of pipeline installation and their placement subject to the rows of plants). Two causes of water loss—evaporation and infiltration below the root-containing zone—along with the need to maintain water availability for plants, should be balanced while designing drip irrigation systems [1,2].

One of the main approaches to the calculation of drip irrigation system parameters is the use of mathematical tools, in particular scenario modelling of processes in the "soil–plant–atmosphere" system, primarily moisture transport processes. The models used for this purpose under the conditions of drip irrigation [3–5] are based mainly on the Richards partial differential equation [6] in a two-dimensional approximation. Their use makes it

possible, without significant changes in the model, to obtain predictive estimates of the distribution of moisture in soil when the irrigation regime, soil, or crop parameters change.

One of the most widely used simulation tools is the HYDRUS-2D software, which models moisture and heat transport in a two-dimensional approximation [7]. HYDRUS-2D uses models such as the van Genuchten–Mualem model [8] and the Brooks–Corey equation [9] to describe the hydro-physical properties of soil. Despite a wide range of features implemented in this software, in some situations there is still a need to develop new modeling tools. Such situations, particularly, include the need to solve inverse problems for decision support while determining the values of SDI system parameters [10–13]; the need to use different models of soils' hydro-physical properties, or to use non-classical moisture transport models (e.g., [14]).

Despite a significant number of case studies and model modifications, one of the more poorly studied aspects is the modelling of moisture transport for crops, particularly potato, grown under raised bed technology. Peculiarities of the raised bed technology are often not taken into account when modelling moisture transport under irrigation (see, for example, [15], where the simulation is carried out in a rectangular domain). Several works, such as [16,17], consider the curvilinearity of the simulation domain, but focus on the study of the influence of some specific factors such as the presence of plastic mulch on the soil surface.

However, using raised bed technology, the main problem in irrigation management is to predict the availability of moisture to plants in the raised beds and adjacent soil zones where root systems are located. Thus, a rarely highlighted aspect regarding drip irrigation is that one of the most effective regimes of its application is the pulse regime of water supply [18,19], the essence of which is the synchronous compensation of crop moisture consumption for transpiration, by the supply of water in short pulses (up to 15 min).

In this context we devoted our study to the assessment of subsurface drip irrigation efficiency in crops growing in raised beds by the modelling of moisture transport, focusing on the accuracy of predictive modelling; the influence of raised bed dynamics on it; and the determination of the values of drip irrigation system parameters that ensure minimal water losses.

## 2. Materials and Methods

For the purpose of modelling, we used the Richards equation [6] stated in terms of water head in a two-dimensional approximation, similar to that presented in [20]:

$$C(h)\frac{\partial H}{\partial t} = \frac{\partial}{\partial x}\left(k(H)\frac{\partial H}{\partial x}\right) + \frac{\partial}{\partial z}\left(k(H)\frac{\partial H}{\partial z}\right) - S,\ 0 \le x \le L_x,\ 0 \le z \le L_z,\ t \ge 0 \qquad (1)$$

where $h(x,z,t) = \frac{P(x,z,t)}{\rho g}$ is the water head $m$; $H(x,z,t) = \frac{P(x,z,t)}{\rho g} + z$ is the full moisture potential, $m$; $P(x,z,t)$ is the suction pressure, $Pa$; $\rho$ is the density of water, $kg/m^3$; $g$ is the acceleration of gravity, $m/s^2$; $C(h) = \frac{\partial \theta}{\partial h}$ is the differential soil moisture content, $\%/m$; $\theta(x,z,t)$ is the volumetric soil moisture content, $\%$; $k(H)$ is the hydraulic conductivity, $m/s$; and $S(x,z,t)$ is the source function, $\%/s$, which models the extraction of moisture by plant roots and its supply by subsurface drip irrigation.

Water retention curves of the soil are represented according to the van Genuchten model [8] in the form

$$\theta(h) = \theta_r + \frac{\theta_s - \theta_r}{\left[1 + (10\alpha|h|)^n\right]^{1-1/n}} \qquad (2)$$

with the values of the coefficients $\theta_r$, $\theta_s$, $\alpha$, $n$ changing from layer to layer. Their values are obtained using the least-squares fitting to the data of experimental studies. The dependency

of the hydraulic conductivity on the water head is represented according to Mualem's model [21] in the form

$$k(h) = k_f \theta_r^\beta(h) \left[ 1 - \left( 1 - \theta_r^{n/(n-1)}(h) \right)^{1-1/n} \right]^2, \quad \theta_r(h) = \frac{\theta(h) - \theta_r}{\theta_s - \theta_r} \tag{3}$$

where $k_f$ is the hydraulic conductivity of saturated soil, and $\beta$ is a fixed exponent. The values of the coefficients here are also determined by fitting them to the experimentally obtained dependencies $k(h)$.

The forms of boundary conditions are given in [20]. They include only gravitational flow condition $\frac{\partial H}{\partial z} = 0$ on the bottom of the domain; symmetric flow conditions $\frac{\partial H}{\partial x} = 0$ on its left and right side; and the condition of flux-controlled interaction with the atmosphere on the upper side. The latter condition has the form

$$k(h, z) \frac{\partial H}{\partial z} = Q_e(t) - Q_p(t) \tag{4}$$

where $Q_e(t)$, $Q_p(t)$ are the fluxes, m/s, of evaporation and precipitation.

The function $S$ models the extraction of moisture by the root systems of plants the way it is described in [20]. The distribution of transpiration along the depth $z$ is described according to [22] in the form

$$S_z(z, t) = \frac{T(t) L(z)}{\int_0^{z_r} L(z) dz} \tag{5}$$

where $z_r$ is the depth of the root-containing layer, $T(t)$ is the transpiration rate, m/s.

In this study, because our experimental analysis included only the determination of root system depth, we used the following form of the function of the distribution of root length density that can be found in [23]:

$$L(z) = 1.44 - 0.14 \frac{z}{z_r} - 0.61 \left( \frac{z}{z_r} \right)^2 - 0.69 \left( \frac{z}{z_r} \right)^3 \tag{6}$$

We model $n_p$ plants (1 or 2 in our computational experiments) with the depth of the root system equal to $r_p$, and centres located in the points $x_{pi}, i = 0, \ldots, n_p - 1$ of the simulation domain. The density of the root system is assumed to decrease linearly subject to the horizontal coordinate $x$ that is described by the function

$$S_{xi}(x) = \begin{cases} \frac{r_{pi} - |x - x_{pi}|}{r_{pi}^2}, & r_{pi} - |x - x_{pi}| \geq 0 \\ 0, & r_{pi} - |x - x_{pi}| < 0 \end{cases} \tag{7}$$

Then the total moisture extraction function has the form

$$S_T(x, z, t) = \frac{1}{n_p} S_z(z, t) \sum_{i=0}^{n_p - 1} S_{xi}(x) \tag{8}$$

To model subsurface drip irrigation, we add to $S_T(x, z, t)$ the density of irrigation water flow $Q_{ss}(x, z, t) = Q_{ss0}(t) \delta(x_{ss}) \delta(z_{ss})$, where $Q_{ss0}(t)$ is flow density from one emitter; 1/s, $x_{ss}, z_{ss}$ are the coordinates of irrigation pipeline location in the simulation domain; and $\delta(\cdot)$ is the Dirac delta function. Finally, we obtain

$$S(x, z, t) = S_T(x, z, t) + Q_{ss}(x, z, t) \tag{9}$$

To subdivide evapotranspiration $ET$ into evaporation flow $Q_e$ (included in the upper boundary condition) and transpiration $T$ (distributed within the root system and included

in the source function $S$), we use the values of LAI (leaf area index) and an empirical parameter $\mu$ [24] as follows:

$$T = M \cdot ET, \, M = 1 - e^{-\mu \cdot LAI} \tag{10}$$

The two-dimensional model based on Equations (1)–(10) assumes that the distance between the emitters is sufficient for the formation of uniform wetting in the plane along the pipeline.

In order to take into account the crop coefficient, evapotranspiration estimation errors and, in general, to adapt the model to the actual growing and soil conditions, a multiplier $k_{ET}$ is introduced to the model [20]; yielding the following representation of evapotranspiration: $ET = k_{ET}ET'$ where $ET'$ is the value of potential evapotranspiration. A similar multiplier $k_{prec}$ is also introduced for the amount of precipitation, to account for possible errors of measurement and boundary condition discretization: $Q_p(t) = k_{prec}Q'_p(t)$ where $Q'_p(t)$ is the measured precipitation. The multiplier $k_{irr}$ for water supply yields $Q_{irr} = k_{irr}Q'_{irr}$ where $Q_{irr}$ is the flow rate of irrigation water from one emitter used in the model, and $Q'_{irr}$ is the corresponding value according to the project documentation of the system. This multiplier allows model adjustment to errors of the Delta function discretization and the decrease in flow rate due to such processes as emitter clogging.

The experimental part of the research was carried out on the lands of the farm "Kyivska" in the village of Makovyshche, Buchansky district, Kyiv region, Ukraine (50.457891 lat., 29.887634 long.). The soil in the research area is grey gilded sandy loess. Coefficients of the van Genuchten-Mualem model for this soil, obtained using the laboratory study [25] data via the minimization of the least squares goal function, are given for two soil layers in Table 1. For the soil in the raised bed, laboratory studies could not be conducted, so the parameter values were set assuming that in the process of its formation the soil loosens and the rate of moisture transport increases.

**Table 1.** Coefficients of the van Genuchten-Mualem model.

| | $\theta r$ | $\theta s$ | $a$ | $n$ | $k_f$, m/s | $\beta$ |
|---|---|---|---|---|---|---|
| In the raised bed | 0.11 | 0.45 | 0.011 | 2 | $8.0 \times 10^{-7}$ | −1.91 |
| Layer 1: 0.1–0.2 m | 0.11 | 0.3655 | 0.011 | 2.3 | $3.97 \times 10^{-7}$ | −1.91 |
| Layer 2: 0.3–0.45 m | 0.04 | 0.3412 | 0.009 | 1.9 | $1.32 \times 10^{-7}$ | −2.64 |

Laboratory-determined and modelled water retention curves are shown in Figure 1. The dependencies between the hydraulic conductivity and the pressure are shown in Figure 2. As can be seen from Figure 2, the Mualem model did not allow describing the laboratory-determined dependencies with high accuracy. When using another model widely applied for Ukrainian soils—the Averyanov model [26]—the accuracy was even lower. In this regard, in addition to the above-described empirical multipliers, in the studied conditions there also was a need to calibrate the form of the dependency between the hydraulic conductivity and the pressure.

The simulation domain is shown in Figure 3.

Initial water head distribution $H_0$ was calculated by iterative smoothing its values with fixed known initial pressures obtained from sensors in given points of the simulation domain [20].

The used numerical modelling technique is based on the finite-difference approximation of Equation (1), and is similar to the one described in [27].

To calibrate the model based on the measurements of pressure values (the scheme of sensor placement is given in Figure 4) during one irrigation cycle, we performed the fitting of the values of its empirical parameters (multipliers $k_{ET}$, $k_{prec}$, and $k_{irr}$), as well as the parameters, the assessment of which is difficult or may not be accurate enough, in particular the parameter $\mu$ used for the separation of evapotranspiration into evaporation and transpiration components, and the soil's saturated hydraulic conductivities.

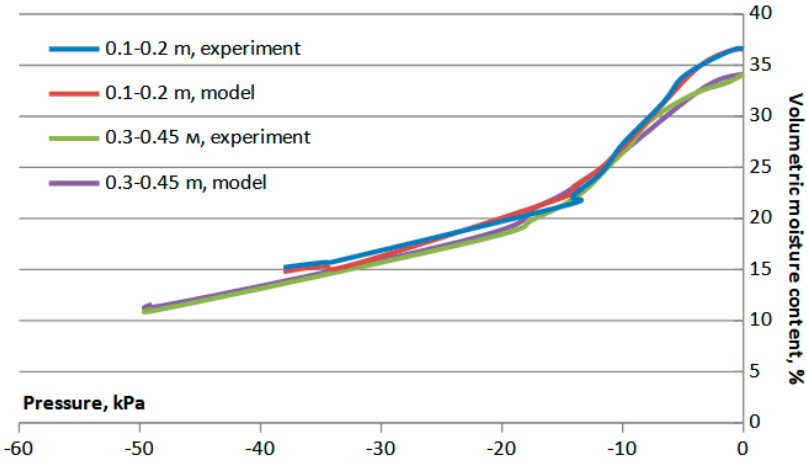

**Figure 1.** Water retention curves of soil layers.

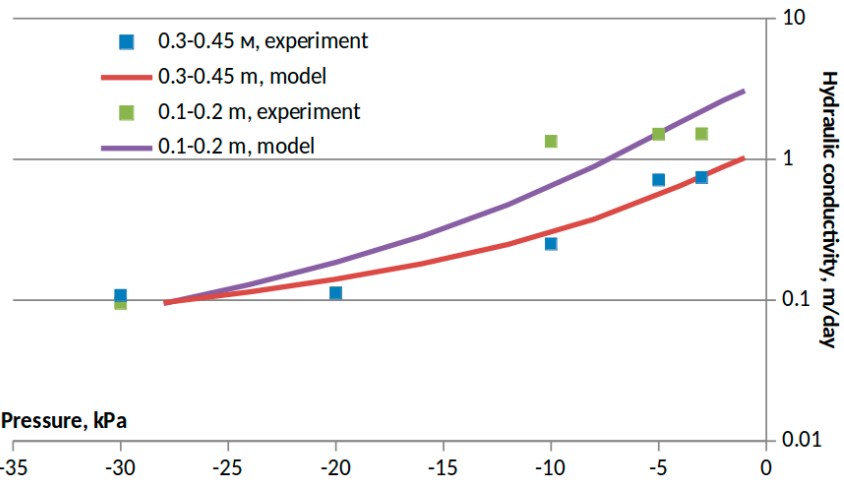

**Figure 2.** Hydraulic conductivity of soil layers.

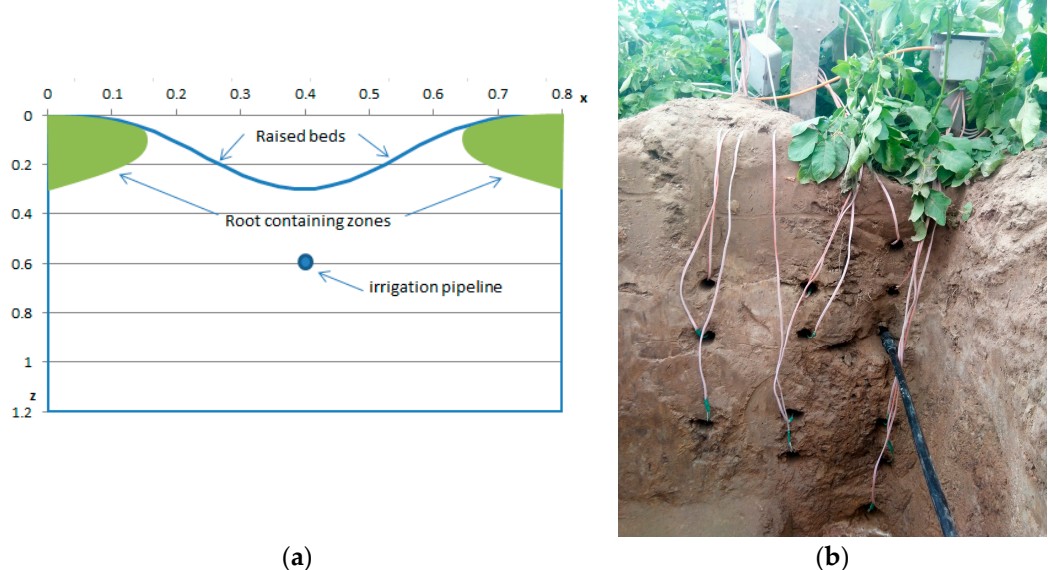

(**a**)                                                    (**b**)

**Figure 3.** Simulation domain (**a**) (raised bed height—30 cm above soil surface level between beds, root system depth—30 cm below raised bed top) and (**b**) actual soil section.

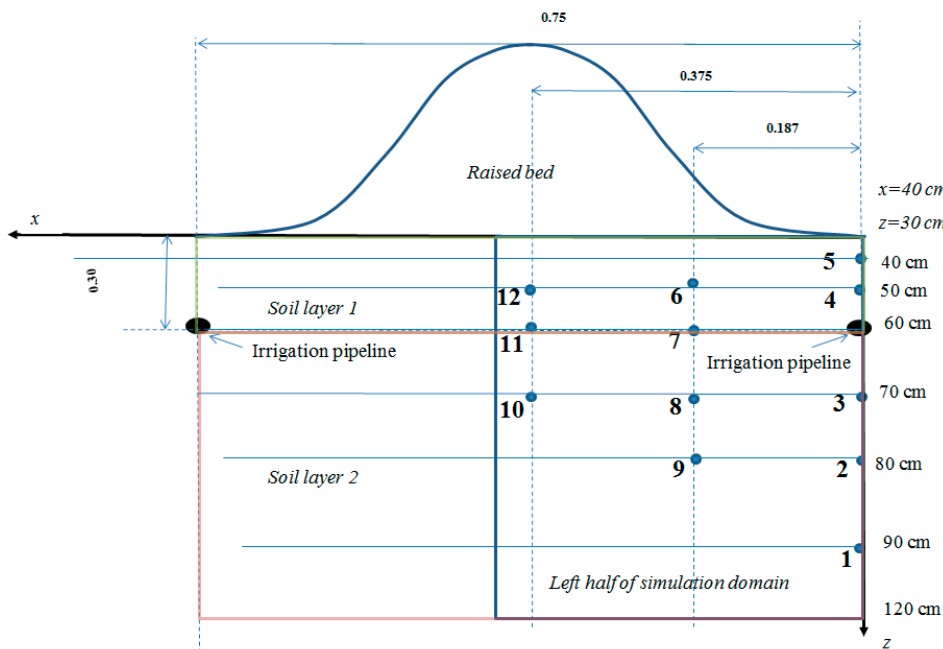

**Figure 4.** Placement scheme for Watermark suction pressure sensors.

We used Watermark sensors for measuring suction pressure.

In the conducted experiment, irrigation pipeline was placed between the raised beds to study the irrigation efficiency of such placement in specific soil conditions. After calibrating the model, we performed scenario modelling with different simulated pipeline placements to find the scenario in which, particularly, the infiltration into deeper layers of soil is minimal.

Calibration was carried out by solving the inverse problem for Equation (1) using the particle swarm optimization (PSO) technique [28]. The calibration method is described in detail in [10,29]. In short, we assumed that there are known values of water head $H_i$ measured at the moments of time $T_i$ in the points $(x_i, z_i), i = 1, \ldots, N$. We searched for the vector of parameter values $\vec{v} = (k_{ET}, k_{irr}, k_{prec}, \mu, \vec{k}_f)$ where $\vec{k}_f$ is the vector of filtration coefficients for the three considered soil layers that minimises the least-squares goal function

$$F(\vec{v}) = \sum_{i=1}^{N} \left( H(x_i, z_i, T_i, \vec{v}) - H_i \right)^2 \tag{11}$$

where $H(x, z, t, \vec{v})$ is the solution of the problem (1)–(10) obtained for the values of parameters from $\vec{v}$. Taking into account the complexity of the problem, and the fixed number and continuity of parameters to be determined, we used the metaheuristic PSO technique for its solution.

Raw data for the simulation was obtained by the iMetos Base weather station located in the field. Three periods were selected for the study, which included both continuous and pulse irrigation. Pulse irrigation is understood here as the irrigation regime under which water is applied in short pulses, (usually with a duration of not more than 15 min), in order to maintain a high range of water availability to plants, adapting the pauses between pulses and their duration to water consumption. In continuous irrigation, larger irrigation rates are applied without pauses. Data on irrigation and precipitation events are given in Table 2. The data on model parameters, in particular evapotranspiration and simulation periods, are given in Table 3. The experimental irrigations were conducted in production conditions to assess modelling accuracy during practical usage of subsurface drip irrigation. Hence, only three controlled waterings separated in time were conducted in different growing stages of potato.

**Table 2.** Simulated irrigation and precipitation events.

| Starting and Ending Time of Event | Event Duration, Minutes | Water Supply Regime/Precipitation | Volume, m³/ha |
|---|---|---|---|
| 17 June 2023 9.30–11.40 | 130 | Continuous watering | 85 |
| 19 June 2023 8.40–9.40 | 60 | Precipitation | 0.00023 |
| 13 July 2023 9:15–14:53 | 338 | Pulse watering (6 events with 15 min duration on average) | 50.83 |
| 13 July 2023 23.00–14 July 2023 4.00 | 300 | Precipitation | 9.16 |
| 11 August 2023 9.45–18.00 | 495 | Pulse watering (8 events with 14 min duration on average) | 70.91 |
| 14 August 2023 11.30–14.35 | 185 | Continuous watering | 110 |

**Table 3.** Periods and parameters of moisture transport modelling.

| Start of Simulated Period | End of Simulated Period | The Depth of Root System, m | The Width of Root System, m | Average Evapotrans-Piration, mm/day |
|---|---|---|---|---|
| 17 June 2023 0:00 | 20 June 2023 0:00 | 0.27 | 0.3 | 7.6 |
| 13 July 2023 0:00 | 14 July 2023 11:00 | 0.3 | 0.3 | 4.9 |
| 10 August 2023 0:00 | 15 August 2023 0:00 | 0.45 | 0.3 | 3.3 |

With the absence of observations in production conditions, the value of LAI was taken to equal to 2.55 (average value reported in [30]). The measured depth of the root system is given in Table 3. The root system distribution function was assumed to have the form given in Equation (6). Distance between emitters was equal to 0.75 m, with their discharge equal to 2 l/h. Potential evapotranspiration was calculated using the Penman–Monteith equation according to the method described in [31]. Its dynamics in the period from 10 August 2023 to 15 August 2023 is illustrated in Figure 5, and its average values for the three simulated periods are given in Table 3. In the period that started 17 June 2023, air temperature changed in the range from 13.6 to 28.8 °C, with changes in air humidity from 29.5 to 100%. In the period that started 13 July 2023, air temperature changed in the range from 14.8 to 29.3 °C, with changes in air humidity from 47.9 to 100%. In the period that started 10 August 2023, air temperature changed in the range from 11.3 to 31.3 °C, with changes in air humidity from 31.5 to 100%.

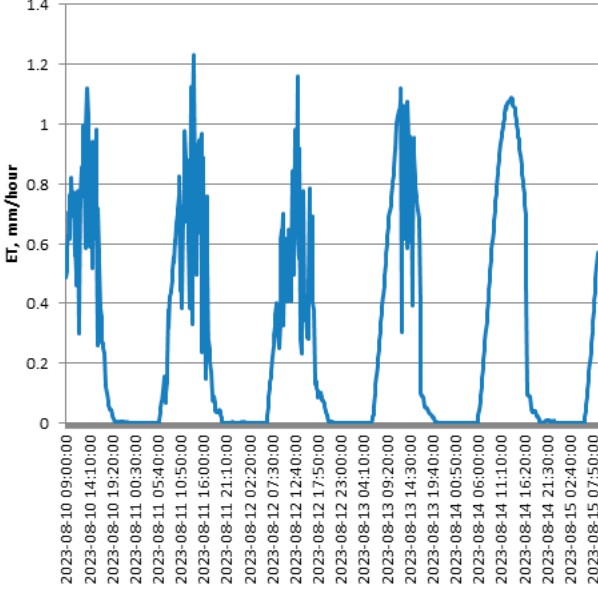

**Figure 5.** Dynamics of potato evapotranspiration for period from 10 August 2023 to 15 August 2023.

The modelling procedure consisted of calibrating the model based on the data collected in the first period, starting on 17 June 2023, after which the accuracy was checked by simulating the water head dynamics in other periods. At the beginning of the third period (10 August 2023), a decrease in the height of the raised beds from 30 to 15 cm was observed. For this period, simulations were carried out both for a height of 30 cm, similarly to the previous periods, and for the observed lower value of height in order to determine the impact of this parameter on the accuracy of simulation.

Further, scenario modeling of watering under different placement of irrigation pipelines was performed.

## 3. Results and Discussion

Model calibration was carried out according to the above-mentioned method, with uniform discretization of the simulation domain with the step with respect to the spatial variables equal to 1.2 cm. The maximum value of the time step was equal to 200 s. The average relative modelling error was calculated as $\varepsilon_1 = \frac{1}{N}\sum_{i=1}^{N}\left|\frac{H(x_i,z_i,T_i,\vec{v}_{best})-H_i}{H_i}\right|$ where we follow the notation of Equation (11), and $\vec{v}_{best}$ is the best found value of parameters vector. In the best case $\varepsilon_1$ was equal to 14.1%. Reducing the sizes of the steps did not lead to a significant change in the error.

The fitted values of the empirical multipliers were as follows: $k_{ET} = 0.93$, $k_{prec} = 7.8$, and $k_{irr} = 3.8$, $\mu = 1.72$. The fitted values of the saturated hydraulic conductivity were equal to $2.39 \times 10^{-6}$ m/s for the raised bed, $2.68 \times 10^{-6}$ m/s for soil layer 1, and $5.42 \times 10^{-7}$ m/s for soil layer 2.

The model needed significant calibration in the considered case, with one of the important sources of errors possibly being the incorrectness of the assumptions regarding processes at the lower boundary of the simulation domain, and low accuracy of the Dirac function's discretization (which led to the high value of the multiplier for irrigation water flux). The other source of errors came from laboratory hydro-physical parameter measurements that were conducted on the extracted and transported soil samples. The large value of the precipitation multiplier could be due to a small amount of precipitation in the training dataset, the measurement of which can have large relative errors.

Thus, these values only make it possible to obtain an approximation of the water head dynamics using a specific discrete model, and cannot be interpreted as certain characteristics of the soil or processes in it.

The average absolute modelling errors $\varepsilon_2 = \sqrt{\frac{1}{N}\sum_{i=1}^{N}\left(H(x_i,z_i,T_i,\vec{v}_{best})-H_i\right)^2}$ for specific sensors both for the range, according to the data collected within which model calibration was carried out, and for other modelling options, are given in Table 4. The average error value among all sensors was 3.16 kPa for the range starting on 17 June 2023; 4.71 kPa for the range starting on 13 July 2023; 5.29 kPa (with a simulated raised bed height of 15 cm); and 5.25 kPa (with a simulated raised bed height of 30 cm) for the range starting on 10 August 2023.

Thus, when calibrating the model on the data collected during continuous watering, the average absolute modelling error $\varepsilon_2$ among all sensors increased by 48% when switching to the pulse regime simulation. However, in our case such an increase can be attributed to the unexpectedly high error for the sensor placed at $x = 0.213$, $z = 0.6$, that could be due to sensor malfunction. Without taking into account this sensor, the error increase is 25%. The change in accuracy when not taking into account the changes in the raised bed height was insignificant compared to the calibration accuracy.

In terms of individual sensors, the error $\varepsilon_2$ decreased with depth, and is in the range from 1.3 to 6.5 kPa for the dataset used for calibration. No significant dependency on the distance to the pipeline was found.

**Table 4.** Average absolute modelling errors, kPa.

| z | x | 17 June 2023 | 13 July 2023 | 8 October 2023 (Raised Bed Height—15 cm) | 8 October 2023 (Raised Bed Height—30 cm) |
|---|---|---|---|---|---|
| 0.1 | 0.4 | 6.5 | 6.2 | 8.2 | 8.2 |
| 0.2 | 0.025 | 3.4 | 4.4 | 8.2 | 8.0 |
| 0.2 | 0.213 | 4.2 | 6.1 | 8.0 | 7.8 |
| 0.2 | 0.4 | 4.0 | 5.0 | 5.0 | 5.0 |
| 0.3 | 0.025 | 1.8 | 2.7 | 6.5 | 6.4 |
| 0.3 | 0.213 | 2.7 | 2.2 | 7.5 | 7.4 |
| 0.5 | 0.025 | 1.6 | 1.7 | 5.6 | 5.7 |
| 0.5 | 0.213 | 1.4 | 5.7 | 4.3 | 4.2 |
| 0.5 | 0.4 | 5.4 | 5.0 | 3.5 | 3.6 |
| 0.6 | 0.213 | 1.6 | 10.7 | 2.1 | 2.1 |
| 0.6 | 0.4 | 4.1 | 3.4 | 2.3 | 2.4 |
| 0.7 | 0.4 | 1.3 | 3.5 | 2.2 | 2.2 |

Despite the fact that not taking into account the change in the raised bed height does not lead to a serious change in the average errors for the sensors placed in the middle of soil massif, the values of the average moisture content of the root layer, which is a critical parameter for irrigation management, differ significantly (Figure 6). This is due to the fact that a large part of the root system was located precisely in the raised bed. As can be seen from Figure 6, with a higher raised bed, irrigation water moistens it longer than in the case of a lower height but further, a higher level of moisture content is maintained in the root zone.

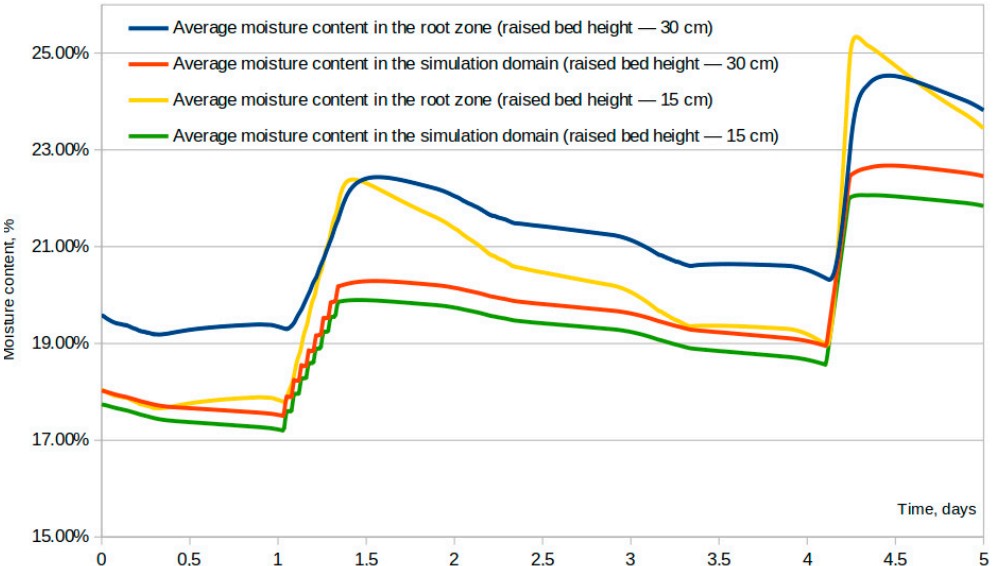

**Figure 6.** The average moisture content in the root zone, and average soil moisture in the simulation domain, for the period from 10 August 2023 to 15 August 2023 for different simulated raised bed heights.

It is worth noting that the simulated dynamics also reflect the process of raising moisture into the root zone at night, when evapotranspiration can tend to zero, with its decrease during the day.

Figures 7 and 8 represent spatial distributions of water heads before and after three of the considered irrigations. Case (a) corresponds to continuous irrigation, while the other cases correspond to pulse irrigation. The obtained results allow making a conclusion that during pulse irrigation, less sharp changes in pressure are formed due to the alternation of water supply and dissipation stages.

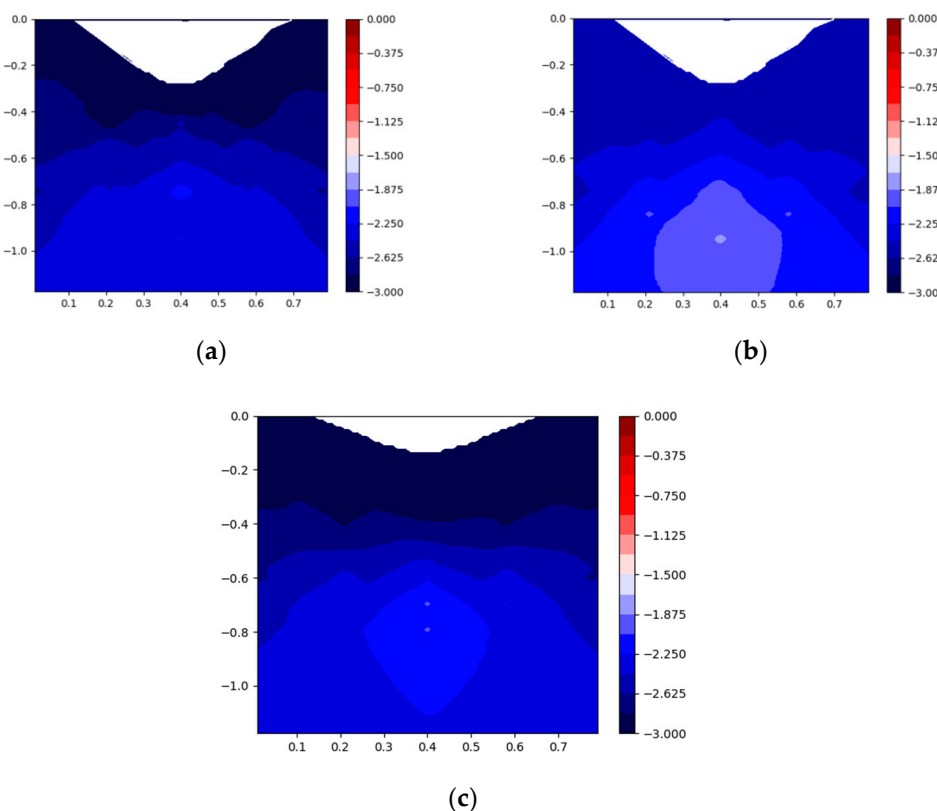

**Figure 7.** Water head, m, distributions before the irrigation on a specific date. (**a**) 17 June 2023; (**b**) 13 July 2023; (**c**) 8 November 2023 (raised bed height—15 cm).

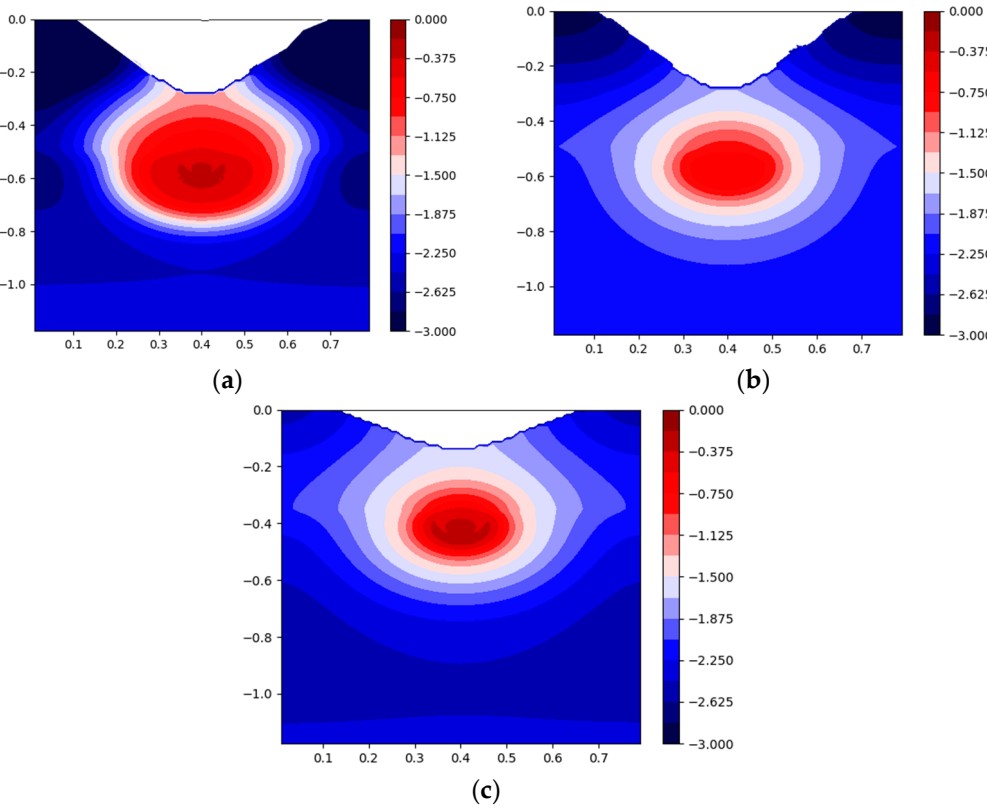

**Figure 8.** Water head, m, distributions at the end of irrigation on a specific date. (**a**) 17 June 2023; (**b**) 13 July 2023; (**c**) 8 November 2023 (raised bed height—15 cm).

Let us note that the considered experimental scenario of pipeline placement between the raised beds is not the only possible option for the organization of subsurface drip irrigation under the raised bed technology of crop growing. Another option is to place the pipeline under the raised bed.

In order to determine the efficiency of irrigation water use, based on the above-described model calibration results, scenario modelling was carried out for different locations of irrigation pipeline, in particular at different depths (10, 20, and 30 cm below the soil level between the raised beds).

The simulation was carried out using the data for the time range from 8 October 2023 to 20 August 2023. Irrigation was simulated when the average suction pressure in the zone where moisture availability is regulated reached $-15$ kPa. Simulated irrigation continued until the average pressure value reached $-5$ kPa. To check the efficiency of the transition from continuous to pulse irrigation regimes we also simulated the case of narrower range modelling irrigation when average pressure reached $-10$ kPa.

Two variants of zones for regulating moisture availability were considered. In the first case, this zone was the entire area of the root system with a depth of 45 cm. Here it was assumed that the moisture regime does not affect the shape of root systems. In the second case, we considered that the highest density of roots is formed in the zone close to the pipeline in the lower parts of raised beds and the upper part of the main soil massif. In the simulation, this zone was located at depths from 7.5 to 22.5 cm below the top of raised beds, the height of which was assumed to be equal to 15 cm. The simulated width of the zone in this case was equal to 50 cm.

Data on the simulated duration of irrigations are given in Table 5. The dynamics of average moisture content in the zone of moisture availability regulation is shown in Figure 9. The dynamics of infiltration flows below the depth of 1 m is shown in Figure 10.

**Table 5.** Duration of waterings, hours, and their number.

| Pipeline Installation Depth, cm | Moistening of Entire Root-Containing Zone | | | | Moistening of Part of Root-Containing Zone Nearest to Pipeline | | | |
| | Pipeline between Raised Beds | | Pipeline under Raised Bed | | Pipeline between Raised Beds | | Pipeline under Raised Bed | |
| | Total Duration | Number of Waterings | Total Duration | Number of Waterings | Total Duration | Number of Waterings | Total Duration | Number of Waterings |
|---|---|---|---|---|---|---|---|---|
| Pre-irrigation threshold $-15$ kPa | | | | | | | | |
| 10 | 12.5 | 3.0 | 14.6 | 6.0 | 11.8 | 3.0 | 13.5 | 5 |
| 20 | 15.0 | 3.0 | 14.2 | 4.0 | 14.3 | 3.0 | 16.0 | 4 |
| 30 | 15.6 | 2.0 | 20.0 | 3.0 | 15.2 | 2.0 | 21.0 | 3 |
| Pre-irrigation threshold $-10$ kPa | | | | | | | | |
| 10 | 17.4 | 6 | 19.4 | 15 | 18.3 | 7 | 19.4 | 13 |
| 20 | 20.6 | 6 | 20.3 | 10 | 19.4 | 6 | 19.5 | 8 |
| 30 | 23.3 | 5 | 25.0 | 6 | 25.2 | 5 | 23.4 | 5 |

According to the simulation results, the availability of moisture to plants is maintained at a given level with minimal irrigation water consumption when placing the pipeline between the raised beds at the depth of 10 cm below the soil level. The reason for this can be the upward movement of water in the night that is more pronounced on the studied soil when the pipeline is placed between the raised beds. This process helps to maintain moisture in the upper layers of soil, increasing its availability for plants and, thus, makes irrigation more efficient.

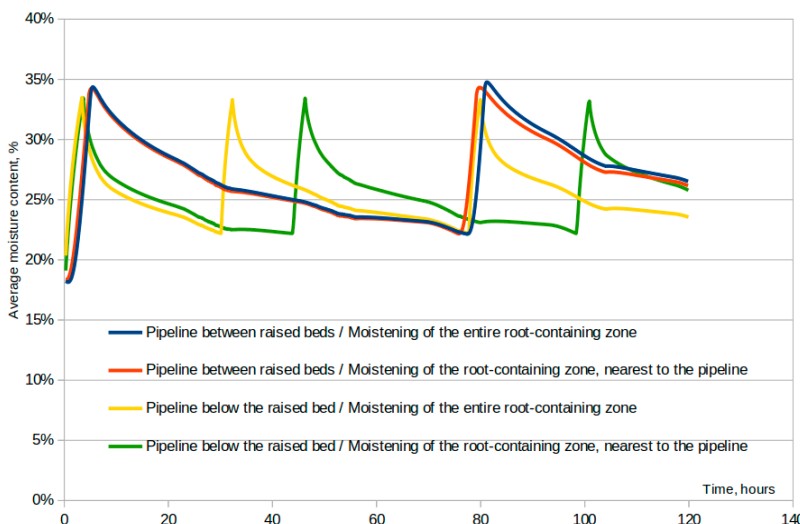

**Figure 9.** Average moisture content in the zone of moisture availability regulation, %, in the case of a pipeline installation depth equal to 10 cm (average initial moisture content in the simulation domain—14%, continuous irrigation).

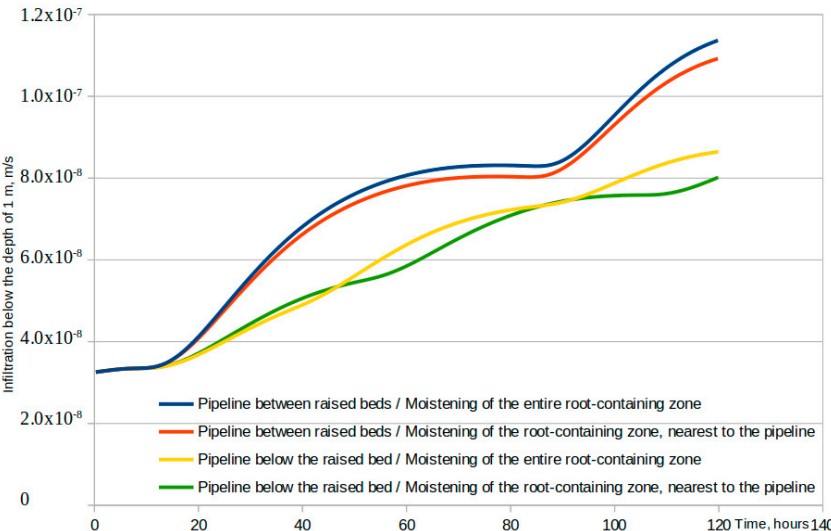

**Figure 10.** Infiltration below the depth of 1 m, m/s, in the case of a pipeline installation depth equal to 10 cm for different horizontal pipeline placements, and the zone of moisture availability maintenance.

Expectedly, for both pre-irrigation thresholds the irrigation amount increased with the increase in pipeline installation depth. When the maintained pressure range is narrower and irrigation duration is shortened, the simulated total irrigation amount increased for the considered soil. This means higher water losses and gives an argument about the inefficiency of pulse irrigation in the case considered.

When installing the pipeline under the raised bed, irrigation should be carried out at a lower rate and with shorter intervals than in the case of installing the pipeline between the raised beds (Figure 9). In the latter case, the interval between irrigations did not depend on the zone in which the availability of moisture to plants is regulated. When placing the pipeline under the raised bed, the interval was shorter in the case when the availability of moisture was regulated in the entire root-containing zone.

Moisture losses due to infiltration below the depth of 1 m (Figure 10) were expectedly greater when placing the pipeline between the raised beds and when regulating the availability of moisture in the entire root-containing zone. They also increase with the increase in pipeline installation depth.

## 4. Conclusions

The conducted simulations show that the considered technique allows the description of the processes of both continuous and pulse water supply regimes with comparable efficiency in a situation where model calibration is carried out for the case of continuous irrigation.

At the same time, the usage of the incorrect values of the raised bed height has a weak effect on the accuracy of modelling inside the soil massif, but significantly changes the simulated dynamics of moisture in the root zone. This proves the fact that while managing irrigation of crops grown under the raised bed technology, it is crucial to monitor changes in raised bed height.

The simulated dynamics reflect such processes as the upper movement of moisture at night into the root-containing zone, located mainly in the raised bed, as well as less sharp changes in the wetting front during pulse irrigation compared to the continuous supply of irrigation water.

The obtained results regarding the calibration of the model allows it to be used in decision-making support systems for irrigated crops, in particular potatoes, using the raised bed technology for their cultivation.

**Author Contributions:** Conceptualization, A.S. and M.R.; methodology, V.B.; software, V.B.; investigation, V.B.; resources, M.S.; data curation, M.S.; writing—original draft preparation, V.B.; writing—review and editing, A.S. and M.R.; project administration, A.S. All authors have read and agreed to the published version of the manuscript.

**Funding:** This research was funded by the Ministry of Education and Science of Ukraine, state registration number of the Grant: 0122U200796.

**Institutional Review Board Statement:** Not applicable.

**Informed Consent Statement:** Not applicable.

**Data Availability Statement:** The data presented in this study are available on request from the corresponding author. The data are not publicly available due to third party restrictions.

**Conflicts of Interest:** The authors declare no conflicts of interest.

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
