# Peer review of "Simulation-Based Assessment of Subsurface Drip Irrigation Efficiency for Crops Grown in Raised Beds"

_2673-4117, doi:10.3390/eng5010024_

Round 1

Reviewer 1 Report (Previous Reviewer 1)

Comments and Suggestions for Authors

This MS now has been greatly improved. after looking at this version, a few comments and questions are raised as followings:

(1) on Fig 4, This dripline displacement may cause much irrigation water moving into deep soil layer where no root develops. this should be discussed in the text. 

(2) LIne 258, It is hard to understand the coefficients of Kprec and Kirr, both have values higher than 1.0. Generally, these values should be less than 1.

(3) Please add a fig to show the water potential distribution before irrigation. with present figure (Fig 7), irrigation water shows little moving to root zone in the raised bed, which in turn influence the irrigation efficiency. 

Author Response

We would like to thank for the comments. According to them several explanations were added to the text.

Answers to the specific comments are the following.

1) on Fig 4, This dripline displacement may cause much irrigation water moving into deep soil layer where no root develops. this should be discussed in the text. 

Answer: In the conducted experiment irrigation pipeline was placed between the raised beds to study irrigation efficiency of such a placement in the specific soil conditions. Further, we performed scenario modeling with simulated different pipeline placements to assess the infiltration into deep soil layers. The appropriate remarks are added in the text.

2) LIne 258, It is hard to understand the coefficients of Kprec and Kirr, both have values higher than 1.0. Generally, these values should be less than 1.

Answer: High value of correction factor for irrigation flow can be explained by the errors induced by the numerical scheme while high value of correction factor for precipitation is due to very low precipitation flow in the dataset used for calibration, which could produce high measurement errors. The corresponding remarks in the text are highlighted.

3) Please add a fig to show the water potential distribution before irrigation. with present figure (Fig 7), irrigation water shows little moving to root zone in the raised bed, which in turn influence the irrigation efficiency. 

Answer: The figures of water potential distribution before irrigation are added.

Reviewer 2 Report (Previous Reviewer 2)

Comments and Suggestions for Authors

1. How to obtain the parameter values in Table 1?

2. The experimental data in Figure 2 is different with last manuscript. Why?

Author Response

We would like to thank for the comments. The answers to them are the following.

1) How to obtain the parameter values in Table 1?

Answer: Parameter values for the considered soil were obtained based on the laboratory study data by the minimization of the least squares goal function. The corresponding remark in the text is highlighted and the reference to the used laboratory study technique is added.

2) The experimental data in Figure 2 is different with last manuscript. Why?

Answer: In the last version of the manuscript the approximation of the experimental data using an exponential function was shown. We changed the representation to show the original measured pairs of pressure and hydraulic conductivity values.

This manuscript is a resubmission of an earlier submission. The following is a list of the peer review reports and author responses from that submission.

Round 1

Reviewer 1 Report

Comments and Suggestions for Authors

This MS developed a model to simulate the soil water distribution under subsurface drip irrigation. After the model is calibrated and validated, the model with the fitted parameters was used to evaluate water distribution and deep leaching under raised bed when drip tape is placed between and under beds. Results in this study could help to design drip playout and field water use. While this MS should be majorly revised following the comments and suggestions:

(1)   This developed model now is unclear and un-numbed. It is better listed related formulas together.

(2)   The experiment also is not well described. The experimental raised bed picture can be presented. The discharge of emitters, the climatic condition, ETo, root development and distribution in the soil bed, LAI of potato, and related items should be described.

(3)   The parameters for evaluating the model precision should be described.

(4)   Line 194, for the three coefficients, KET, the crop coefficient looks too high for potato, what's the meaning of Kprec and Kirr.

(5)   The table 5 should be rearranged, the present table is hard to understand.

(6)   There are comma between digital numbers in table 4 and 5. Clarify it.

(7)    On Fig 8, The irrigation amount and initial soil moisture conditions, pipeline depth, pulse and continuous irrigation should be described in the figure.

(8)   On fig 9, infiltration rate at the 1 m depth depends on irrigation depth. this should be described in the Figure title

Other comments can be found in the marked file.

Comments on the Quality of English Language

This MS developed a model to simulate the soil water distribution under subsurface drip irrigation. After the model is calibrated and validated, the model with the fitted parameters was used to evaluate water distribution and deep leaching under raised bed when drip tape is placed between and under beds. Results in this study could help to design drip playout and field water use. While this MS should be majorly revised following the comments and suggestions:

(1)   This developed model now is unclear and un-numbed. It is better listed related formulas together.

(2)   The experiment also is not well described. The experimental raised bed picture can be presented. The discharge of emitters, the climatic condition, ETo, root development and distribution in the soil bed, LAI of potato, and related items should be described.

(3)   The parameters for evaluating the model precision should be described.

(4)   Line 194, for the three coefficients, KET, the crop coefficient looks too high for potato, what's the meaning of Kprec and Kirr.

(5)   The table 5 should be rearranged, the present table is hard to understand.

(6)   There are comma between digital numbers in table 4 and 5. Clarify it.

(7)    On Fig 8, The irrigation amount and initial soil moisture conditions, pipeline depth, pulse and continuous irrigation should be described in the figure.

(8)   On fig 9, infiltration rate at the 1 m depth depends on irrigation depth. this should be described in the Figure title

Other comments can be found in the marked file.

Reviewer 2 Report

Comments and Suggestions for Authors

1. The simulation accuracy of Hydraulic conductivity of soil layers is very low in Figure 2. 

2. The experimental arrangement is not clearly described. Where did the depth of ridge formation and drip irrigation pipes start to be calculated in Figure 3? What is the dripper flow rate? How long does each irrigation or precipitation last?

3. What was the relationship between the Figure 3 (raised beds) and Figure 4 (horizontal ground)?

4. What did  “9, 16” and “70, 91” in Table 2 mean? What did “6 x 15 min” mean?

5. Line 151:  What sensors were used for measuring the pressure values? Accuracy?

6. How to calibrate the model?

7. Which area in Figure 3 are the root zone and the simulation domain in Figure 6?

8.Figure 7 should analysis soil moisture content that be consistent with Figure 6.